# Physical Exercise Induces Significant Changes in Immunoglobulin G N-Glycan Composition in a Previously Inactive, Overweight Population

**DOI:** 10.3390/biom13050762

**Published:** 2023-04-27

**Authors:** Nina Šimunić-Briški, Robert Zekić, Vedran Dukarić, Mateja Očić, Azra Frkatović-Hodžić, Helena Deriš, Gordan Lauc, Damir Knjaz

**Affiliations:** 1Genos Ltd., 10000 Zagreb, Croatia; 2Faculty of Kinesiology, University of Zagreb, 10000 Zagreb, Croatia; 3Faculty of Pharmacy and Biochemistry, University of Zagreb, 10000 Zagreb, Croatia

**Keywords:** IgG, glycosylation, N-glycans, inflammation, exercise, sedentary population

## Abstract

Regular exercise improves health, modulating the immune system and impacting inflammatory status. Immunoglobulin G (IgG) N-glycosylation reflects changes in inflammatory status; thus, we investigated the impact of regular exercise on overall inflammatory status by monitoring IgG N-glycosylation in a previously inactive, middle-aged, overweight and obese population (50.30 ± 9.23 years, BMI 30.57 ± 4.81). Study participants (*N* = 397) underwent one of three different exercise programs lasting three months with blood samples collected at baseline and at the end of intervention. After chromatographically profiling IgG N-glycans, linear mixed models with age and sex adjustment were used to investigate exercise effects on IgG glycosylation. Exercise intervention induced significant changes in IgG N-glycome composition. We observed an increase in agalactosylated, monogalctosylated, asialylated and core-fucosylated N-glycans (padj = 1.00 × 10^−4^, 2.41 × 10^−25^, 1.51 × 10^−21^ and 3.38 × 10^−30^, respectively) and a decrease in digalactosylated, mono- and di-sialylated N-glycans (padj = 4.93 × 10^−12^, 7.61 × 10^−9^ and 1.09 × 10^−28^, respectively). We also observed a significant increase in GP9 (glycan structure FA2[3]G1, β = 0.126, padj = 2.05 × 10^−16^), previously reported to have a protective cardiovascular role in women, highlighting the importance of regular exercise for cardiovascular health. Other alterations in IgG N-glycosylation reflect an increased pro-inflammatory IgG potential, expected in a previously inactive and overweight population, where metabolic remodeling is in the early stages due to exercise introduction.

## 1. Introduction

The World Health Organization considers high blood pressure, smoking, high blood sugar, physical inactivity, obesity, high cholesterol, risky unprotected sex and alcohol consumption to be the main risk factors for death [1]. Based on a larger number of studies [2,3,4,5,6,7], it can be concluded that physical inactivity and a sedentary lifestyle lead to health problems manifesting in excess body weight. The American College of Sports Medicine (ACSM) recommends a minimum of 150 min per week spent in the field of cardiorespiratory activities, i.e., five times per week for 30 min [8]. To develop muscle strength, it is necessary to participate 2–3 times a week in training with an external load. Maximal oxygen uptake is defined as the capacity of the cardiorespiratory system to supply blood to the working skeletal muscles. Aerobic activities in the range of 40–59% of V02max and 55–70% of maximum heart rate (HRmax) are considered moderate, while high intensity falls in the range of 60–84% of V02max and 70–89% of HRmax. It is well known that regular exercise over a longer period of life has a positive impact on many aspects of human health.

Musculoskeletal, cognitive and metabolic functions all benefit from regular exercise, while risks of some of the leading diseases with the highest overall mortality rates worldwide such as cardiovascular diseases, diabetes mellitus type 2 and some types of cancer can be reduced by introducing regular movement into everyday life [9,10]. Cardiorespiratory fitness (CRF) is considered one of the most important measures of health, measured by V02max. Even a one metabolic equivalent (1 MET) increase in an exercise test showed 12% improvement in survival among men suffering from cardiovascular disease (CVD), showing that V02max is one of the most powerful predictors of mortality [11,12]. A sedentary lifestyle speeds secondary aging of V02max by 30 years [10]; from the age of 20 to 70, V02max will decrease by 40% [13]. Onset of type-2 diabetes (T2D) can, among other factors, be modulated by lifestyle choices; regular exercise is one of the powerful factors, as a single exercise bout increases glucose uptake of the skeletal muscle, thus lowering insulin resistance in T2D patients [14,15].

Regular exercise is associated with a lower number of senescent T-cells, increased T-cell proliferation and neutrophil phagocytic activity, greater NK-cell cytotoxic activity as well as lower inflammatory response in bacterial infections and overall lower circulatory levels of inflammatory cytokines [16]. When exercise is continuous and lasts over 90 min, being performed at moderate to high intensity at 55–75% of aerobic capacity, without sufficient food intake, post-exercise immune depression occurs. Usually, within 24 h, immune function is recovered [17]. Immune suppression in athletes is still a debate, some research showing little to no effect of high-intensity training on immune function [18], while some reach different conclusions. Many studies have shown that both pro- and anti-inflammatory cytokines (IL-1ra, IL-6, IL-8 and IL-10) increase following longer endurance exercise, but their response is far less significant after short-duration intensive exercise, showing that cytokine responses are more related to exercise intensity than dependent on exercise-induced muscle damage [19,20,21,22].

Immunoglobulin G (IgG) is the most abundant antibody in the human body, found in all body fluids, secreted by plasma cells and involved in several humoral immune processes: antigen neutralization, complement activation, complement dependent cytotoxicity (CDC) and antibody-dependent cell-mediated cytotoxicity (ADCC), as well as hypersensitivity reactions [23]. Most of the known proteome contains different types of glycans, which are highly responsive to both environmental as well as genetic stimuli. The environment has a strong impact on glycans, allowing them to frequently change depending on various physiological conditions [24]. The IgG glycome denotes all the glycans present in an individual’s IgG molecules. Glycans make up to 15% of IgG weight, with significant influence on folding, stability, function and structure. Glycan removal results in the loss of both the anti- and pro-inflammatory activity of IgG. Glycans regulate the cellular and humoral immune responses, including assembly of peptide-loaded major histocompatibility complex (MHC) antigens, reorganization of T-cell receptor complexes, modulation of immune receptor clustering, endocytosis, receptor signaling and immunoglobulin functions [25]. Francesci et al. (2018) showed that changes in IgG N-glycome influence inflammatory response [26]. Changes in IgG glycome occur with aging or with the onset of different types of diseases. Age-related diseases are types of diseases whose prevalence increases with age: mainly CVDs, T2D, osteoporosis and types of cancer. Changes in N-glycosylation and in N-glycan composition have been observed in all major subtypes of diabetes [27], as well as in CVDs where N-glycosylation alterations are associated with atherosclerosis development [28]. As IgG glycome is only partially heritable, the influence of the environment accounts for the composition as well, so lifestyle choices, chronological age and possible diseases count as variables to address. IgG glycans are considered as excellent biomarkers of overall health and can represent an individual’s biological age. A balance between anti- and pro-inflammatory glycans of IgG is extremely important for healthy aging [29]. Immunosenescence is explained as a loss of balance between inflammatory and anti-inflammatory networks, resulting in a low-grade chronic pro-inflammatory status, called ‘inflamm-aging’. Many age-related diseases have a similar inflammatory pathogenesis, but long-lived individuals seem to have managed to counteract subclinical inflammation through an anti-inflammatory counterresponse maintaining a delicate balance [30]. The term ‘inflamm-inactivity’ was introduced most recently, determining the proportion of inflamm-aging related to the sedentary lifestyle that seems to increase with age [31], as physical exercise has a positive effect on many of the beforementioned chronic diseases. Due to a high level of body fat, usually accompanied with less and less physical activity, inflammation is known to increase in the later stages of life. Several studies have linked obesity with pro-inflammatory IgG N-linked glycans, confirming the association of higher central adiposity levels with an increased pro-inflammatory fraction of IgG glycans [32]. An intervention study resulting in significant weight loss in obese participants revealed a significant shift from pro- to anti-inflammatory activity of IgG [33,34].

Very few studies have investigated the effect of exercise on IgG N-glycosylation. Investigation of several molecular pathways, including IgG N-glycosylation, studied immunosuppression in healthy females undergoing a prolonged training period of 18 weeks of intense training with low energy intake, observing an IgG alteration shift towards pro-inflammatory activity, although body composition revealed significant fat loss [35]. Another study investigated the inflammatory effects of anaerobic all-out exercise in healthy young males, reporting an IgG shift towards pro-inflammatory directly after 4 weeks of exercise and shifting back to an anti-inflammatory profile after the recovery period [36]. As low-level inflammation occurs after every exercise bout due to the micro-injuries caused to the working muscle, the impact of inflammation on the overall status of individuals over longer periods of time is an interesting subject to investigate further, as well as the overall impact of regular exercise in previously sedentary older young adults and younger middle-age population.

This study aimed to investigate the impact of regular exercise on overall inflammatory status by monitoring IgG N-glycosylation in a previously inactive, middle-aged, overweight and obese population, as physical activity is one of the main non-invasive approaches in the treatment of obesity, which is associated with poor metabolic health and various diseases.

## 2. Materials and Methods

### 2.1. Participants

The study population included 397 apparently healthy, but previously physically inactive, people (F = 313 (78.84%), M = 84 (21.16%); average age 50.30 ± 9.23 years; average height 168.46 ± 8.96 cm; average weight 87.09 ± 17.20 kg; body fat 39.76 ± 8.30%, BMI 30.57 ± 4.81 kg/m^2^). Physical inactivity was determined by a questionnaire on the general state of health and related to the last 5 years, during which the respondents were not actively involved in any form of recreational activity. The questionnaire consisted of questions related to possible locomotor issues, chronic diseases and medications (medical history) and questions aimed at determining the level of physical activity during the last 5 years. The initial testing confirmed that the subjects did not have comorbidities related to locomotor system difficulties and other chronic diseases that could prevent them from normal participation in the training process adapted to the capabilities and needs of the tested sedentary population. Taking into account the responses of the participants in the questionnaire itself and the results achieved during the initial testing, the group of kinesiologists responsible for the implementation of training activities made a selection for a particular type of training program, in agreement with a medical doctor.

A total of 450 respondents were selected to participate in the training programs and were tested initially, whereas 397 participants finished the whole training program and participated in the final testing (drop-out rate = 13.35%).

### 2.2. Experimental Protocol

All subjects underwent the initial and final testing, which included the collection of dry blood spot (DBS) samples, determining success level in basic motor tests and the assessment of functional abilities. Basic motor tests encompassed alternate lunges (number of repetitions in 30 s), plank (measured in seconds), incline push-ups (number of repetitions in 60 s), squats (number of repetitions in 60 s) and wall sit (performed until termination). Assessment of functional abilities included a modified beep test (15 m distance of sections).

Dry blood spots were collected from all participants at baseline and at the end of the intervention. The selected finger was wiped with sterile medical ethanol and allowed to dry. Skin puncturing was performed with a 21 G BD contact-activated lancet (BD, Franklin Lakes, NJ, USA) allowing for the blood drop to form. The blood spot was collected on Whatman Human ID Bloodstain Cards, Cat. No. WB100014 (GE Healthcare, Anaheim, CA, USA). Cards containing blood spots were left to dry at room temperature for 2 h and were then stored in airtight zipped bags with desiccant at −20 °C.

After conducting the initial testing, the subjects were included in one of the three exercise programs under the supervision of an expert kinesiologist. The 12-week exercise programs 2× per week, 60 min per exercise, included a circular exercise program (*n* = 229), a cardio exercise program (*n* = 115) and a Nordic walking program (*n* = 53). The participants were randomly divided into smaller groups of 15–20 participants. Each group of participants underwent an assigned exercise program in the course of 12 weeks. Each program was adapted based on the initial state of morphological characteristics and motor and functional abilities. After the end of the 12-week program, the subjects underwent final testing, which was conducted according to the same protocol as the initial testing procedure.

In accordance with the recommendations, the implemented plan and program of systematic training in this research were carried out as follows: in one program, exercises were dominantly focused on the development of the cardiorespiratory system; the second program consisted of circular exercises with an emphasis on the activation of all major muscle groups with an external load or the weight of one’s own body; the third program, Nordic walking, focused on an aerobic component, but at the same time, the activation of larger muscle groups of the upper and lower extremities was present.

#### 2.2.1. Circular Exercise Program

The circular work program meant performing exercises in a circular form, where the work period and the rest period alternated, and the exercise itself was carried out in several rounds between which there was a longer rest period. The exercises were performed in a specific order, with the aim of activating all major muscle groups. The training was most often programmed in such a way that the activation of the topological regions of the body alternated. The exercises were performed with an external load such as dumbbells, bands and other available props. The intensity of exercise progressively increased, and accordingly, the exercises were performed from simpler to more complex.

#### 2.2.2. Cardio Exercise Program

The cardio work program meant exercising on trainers in the gym (e.g., treadmill, bike ergometer, rowing ergometer and elliptical), where work and rest intervals alternated, predominantly dominated by the work component. The intensity and extent of the training load increased progressively, and each training period was aimed at training on the provided trainer or more, with the aim of reaching a certain training zone.

#### 2.2.3. Nordic Walking Program

Nordic walking training activated various muscle groups of the upper and lower body. The use of poles made it possible to increase the intensity of exercise without having a negative impact on the musculoskeletal system of previously physically inactive people. The intensity and extent of the training load increased progressively, and the goal of each training session was to exercise in the aerobic-extensive zone.

#### 2.2.4. N-Glycan Analysis from DBS

IgG N-glycan analysis was performed following the protocol described in detail elsewhere [37], with some minor modifications. First, DBS were cut into smaller pieces that were placed in a 96-well collection plate and gently shaken for three hours at room temperature with 800 μL 1x phosphate buffer saline (1× PBS, prepared in-house). IgG from DBS was isolated using protein G monolithic 96-well plates (BIA separations, Ajdovščina, Slovenia) following an optimized high-throughput method [38]. Briefly, IgG was eluted with 0.1 M formic acid (prepared in-house) and rapidly neutralized with 1 M ammonium bicarbonate (also prepared in-house). IgG eluate (600 μL) was dried in a vacuum concentrator overnight. Dried IgG was then desalted with 800 μL methanol (Honeywell, Charlotte, NC, USA) cooled down to −20 °C. Samples were resuspended and centrifuged at 2000× *g* for 15 min, and 78 μL of the supernatant was carefully removed. This step was repeated by adding another 800 μL of cold methanol to the protein precipitate. Samples were resuspended and centrifuged, and finally, the protein precipitate was left to dry in a vacuum concentrator for two hours [37]. N-glycans were enzymatically released from dried and desalted IgG with a specific enzyme, PNGase F, and free IgG N-glycans were then fluorescently labelled with 2-aminobenzamide (2-AB) [39]. Fluorescently 2-AB labelled IgG N-glycans were analyzed by hydrophilic interaction liquid chromatography (HILIC) ultra-performance liquid chromatography (UPLC), as described previously by Trbojević et al. [39]. The obtained chromatograms were manually integrated and separated into 24 glycan peaks. To make the measurements comparable across the samples, normalization was performed by expressing the amount of glycans in each peak as a percentage of the total integrated area. A list of IgG N-glycan structures corresponding to individual glycan peaks is available in Appendix A.

### 2.3. Statistical Analysis

Glycan measurements were log-transformed, and batch correction was performed using the ComBat() function in the “sva” package in R, where a plate was modelled as a batch covariate. Upon subtraction of estimated batch effects from log-transformed values, the values were exponentiated to obtain corrected measurements on the original scale. After normalization and batch correction, eight derived glycan traits were calculated, each representing a percentage of structurally similar glycan species in the total IgG glycome. Formulae used for calculation of derived traits can be found in Appendix A. The analysis of the effect of exercise during the 3 months on glycan levels was performed by implementation of a linear mixed model. The analysis was performed in the whole sample and exercise-specific subgroups (circular, cardio and Nordic walking). Glycan values (direct glycan values and derived glycan traits) were modelled as dependent variables, and time was modelled as a fixed effect, while participant ID was included as a random intercept, with age and gender included as additional covariates. An inverse transformation of ranks to normality was applied to glycan traits using the rntransform() function in the “GenABEL” R package. A multiple testing correction was performed using the p.adjust() function in R, which implements the Benjamini–Hochberg procedure for false-discovery rate control (method = “fdr”) with an adjusted *p*-value of < 0.05 considered as significant. Data analysis and visualization were performed using R software (version 3.5.1).

Descriptive statistics was used to determine the basic parameters of test results for each group (training programs) in initial and final testing (mean—$\;\bar{x}$; standard deviation—SD). A mixed-model (2 × 3) ANOVA was used to determine interactions and the influence of the training program on test results. The partial ŋ2 coefficient was used as an indicator of effect size. Furthermore, the Tukey post hoc test was performed on variables with significant time*group interactions to determine differences. Statistical analysis of functional and morphological changes as well as changes of motoric skills was performed with the use of Statistica 14.0.1.25 (TIBCO software, Inc., Palo Alto, CA, USA). The level of statistical significance was set at *p* < 0.05.

## 3. Results

### 3.1. Effect of Medium-Intensity Exercise on Motor and Functional Abilities

Basic descriptive parameters for each training program in initial and final measurements are presented in Table 1. Additionally, changes in morphological characteristics were observed (muscle mass and BF%). It can be concluded that after the training program, subjects had a positive improvement in BF%, with BF reduced by 1.89%. Furthermore, it is observed that the influence of all training programs on muscle mass induced a marginal improvement of 0.20 kg. Average results in all tests improved on final testing. Results of the group that participated in circular training had the best average results in five out of six tests. Significant changes were determined in tests: lunges (F = 3.80; *p* = 0.02), squats (F = 8.03; *p* < 0.01), push-ups (F = 7.22; *p* < 0.01) and plank (F = 9.82; *p* < 0.01). In the wall sit and beep test, differences were not significant. Furthermore, for variables with significant differences, a Tukey post hoc test was performed (Table 2).

The post hoc test for significant interactions presented in Table 2 shows that significant differences were determined between initial and final testing of motor abilities in cardio, circular and Nordic walking training programs (*p* < 0.01). Initially, there was no differences between the results of training programs. Moreover, in the final testing of push-ups between cardio and circular training, a difference is observed (*p* < 0.01). Furthermore, the plank test showed differences between Nordic and circular training on final testing (*p* < 0.01).

### 3.2. Effect of Medium-Intensity Exercise on IgG N-Glycosylation

The N-glycome composition of IgG isolated from study participants was evaluated at baseline and after the exercise intervention and was compared between these two timepoints.

Exercise induced significant changes in IgG N-glycome composition when all exercise programs were pooled together (Table 3; Section 1 (all)). It included an increase in agalactosylated, monogalctosylated, core-fucosylated and asialylated N-glycans (β = 0.04, padj = 1.00 × 10^−4^; β = 0.25, padj = 2.41 × 10^−25^; β = 0.51, padj = 3.38 × 10^−30^; and β = 0.25, padj = 1.51 × 10^−21^, respectively), as well as a decrease in digalactosylated, monosialylated and disialylated N-glycans (β = −0.10, padj = 4.93 × 10^−12^; β = −0.09, padj = 7.61 × 10^−9^; and β = −0.53, padj = 1.09 × 10^−28^, respectively). The same effects were observed when participants were divided on the basis of the exercise program. In the circular and cardio programs, significant glycan changes were the same as in the pooled group, while the Nordic walking program showed the same direction of changes for all glycans but lacked statistical significance for some glycans. The complete list of investigated associations between exercise and IgG glycome composition is available in Appendix A.

Figure 1 shows changes in IgG glycosylation-derived traits comparing three months of exercise to the baseline for all the participants, divided into individual groups that differ in their exercise program.

Monogalactosylated glycan with galactose on the 3 arm (FA2[3]G1), labelled as GP9 in the IgG glycome, was recently reported to be an independent protective biomarker for future cardiovascular events in women [40]; thus, we also included this individual glycan in the analysis. Calculated effects of exercise on individual IgG N-glycans showed a significant change in GP9 (β = 0.126, padj = 2.05 × 10^−16^) (Appendix A), as well as for the circular and cardio exercise programs, when stratified for sex, as presented in Table 4. In the Nordic walking program, the effect of exercise follows the same trend as in the circular and cardio programs, but it is not statistically significant, possibly due to the lower number of individuals participating in that program (Figure 2). No significant changes in IgG glycosylation were observed between different exercise programs.

## 4. Discussion

Immunoglobulin G N-glycosylation is known to reflect pro- and anti-inflammatory status, as well as various pathophysiological processes in the organism. However, the effects of regular exercise on IgG glycosylation in a previously inactive, sedentary and overweight population have so far never been investigated. The aim of this study was to determine the impact of a three-month exercise regime on IgG glycosylation and motor and functional abilities in such an at-risk population.

The main result of this study is the positive effect of exercise reflected through a small but measurable increase in FA2[3]G1 IgG glycan (GP9), which was previously reported to have a protective role in cardiovascular health in women [40,41]. Cardio protective GP9 has increased in all three exercise groups, in women as well as in men, although for men GP9 does not have the same cardioprotective indication. Wittenbecher et al. investigated plasma N-glycans as emerging biomarkers of cardiometabolic risk in the EPIC-Potsdam Cohort and reported that FA2[3]G1 (plasma protein GP5 corresponding to IgG GP9) is particularly associated with a lower risk of incident cardiovascular events (myocardial infarction and stroke) in women [41]. A subsequent study, investigating IgG glycosylation in the same population, showed that IgG GP9 is again inversely associated with CVD incidence in women [40].

Next, herein we report a mild pro-inflammatory glycan shift after an initial three months of physical activity in a previously sedentary population. We observed a decrease in digalactosylated, monosialylated and disialylated structures, as well as an increase in agalactosylated, asialylated and core-fucosylated structures, usually associated with the increased pro-inflammatory potential of IgG. This is somewhat to be expected, since our study included an inactive and sedentary population with higher BMI, who had just started the lifestyle intervention. After years of sedentary lifestyle and without diet intervention, it is to be expected that metabolic remodeling is starting slowly. Our results are in accordance with previously conducted studies of exercise impact on IgG glycosylation which reported a shift towards a pro-inflammatory IgG glycan profile in both young male athletes [36] and young females who underwent an energy-deprivation program paired with an intense exercise program in order to induce intensive weight loss [35]. Several studies have tackled the relation between obesity and IgG N-glycans, in order to try to determine the association between increased body fat and IgG glycosylation features. One study reported that central adiposity is in fact related with increased pro-inflammatory IgG glycan structures [32], while the other went one step further, investigating the effect of weight-loss intervention in 37 obese patients, who were first subjected to a low-calorie diet and had undergone bariatric surgery. This study showed that the low-calorie diet induced a decrease in the bisecting GlcNac levels, whose higher levels are associated with ageing and inflammatory conditions. Bariatric surgery that followed, accompanied by continuous weight loss during a one-year follow-up, produced substantial alterations of IgG N-glycome: an increase of digalactosylated and sialylated glycans and a decrease of agalactosylated and core-fucosylated IgG N-glycans, reflecting an enhanced anti-inflammatory IgG potential [33]. Lastly, Deris et al. investigated dietary intervention performed in the DIOGenes study, in order to explore weight-loss-induced changes of IgG glycosylation and follow-up weight-maintenance diets. Again, the most significant IgG N-glycome alterations occurred in the intense weight-loss period induced by a low-calorie diet, where decrease of agalactosylated and increase of sialylated N-glycan structures was reported, indicating a shift from pro- to anti-inflammatory activity of IgG. Follow-up weight-maintenance diets did not show statistically significant changes between diet types, suggesting that calorie intake and, more importantly, weight loss rather than a diet type, are the main drivers of IgG glycome changes [34]. Our population is overweight and, to some extent, obese, with average body fat of 40.71 ± 7.97% and BMI 30.57 ± 4.81 kg/m^2^, which can explain why we are reporting a slightly higher pro-inflammatory shift, as adipose tissue produces adipokines such as IL-6 and TNF-α and deposits them. After years of inactivity and sedentary lifestyle, it well expected that introducing regular exercise into daily life will induce metabolic and immune changes. Our study participants were strictly instructed not to change anything in their dietary habits, and thus the weight loss is minimum, but we observed a slight conversion between composition of body fat versus muscle ratio. We hypothesize that our population is probably undergoing the first stages of metabolic remodeling and the initiation of fat loss due to regular physical exercise, so the observed pro-inflammatory profile of IgG could stem from pro-inflammatory cytokines regularly stored in a body fat compartment, which is now perturbed due to the exercise intervention. Tijardović et al. reported significantly increased pro-inflammatory markers just after exercise intervention in their population, stating that IL-6 and total leukocytes were elevated [36]. Sarin et al. proposed that prolonged exposure to low energy availability paired with extensive exercise may predispose immunosuppression through inhibition of B-lymphocyte maturation and proliferation, which is directly accompanied by reduced IgG levels and IgG glycosylation modulation [35]. In addition, lower levels of galactosylation (monogalctosylated and digalactosylated N-glycan structures) and sialylation (monosialylated and disialylated N-glycan structures) as well as higher levels of core fucosylation have been associated with poorer metabolic health [42]. As with different weight-maintenance diets, we also report there is no ‘one type fits all’, referring to exercise regimens [34].

Exercise-induced immune suppression is a hot topic in exercise immunology. One side reports that there is no negative impact of exercise bouts on immunity [18], while the other side reports post-exercise immune function depression [17], or that anti-inflammatory response is physical fitness-dependent [19]. Simpson at al. admitted that effects of exercise training on the frequency and function of B-cells in the aging population have been neglected, since most of the studies focus on T-cells [16]. Here we would like to offer additional insight on the possible impact of exercise on IgG glycans, as immunoglobulins are produced by B-cells. As mentioned previously, IgG is involved in multiple humoral immune processes including ADCC, a type of immune reaction in which a target cell or microbe is coated with antibodies, which triggers effector cells to induce target cell death via non-phagocytic mechanisms [43]. It has been reported that the absence of the core fucose from IgG1 N-glycan enhances ADCC activity by up to 50–100-fold [44]. Biophysical studies showed that afucosylated antibody binds to the FcγRIIIa receptor with higher affinity than the fucosylated version [45]. However, only afucosylation of antigen-specific antibodies promotes ADCC, while afucosylation of antibodies directed to other antigens may actually suppress ADCC by competition to binding to Fc gamma receptor IIIa. In this research we report elevated levels of core fucose, but further research is needed to understand functional implications of this observation and its impact on the ADCC. Additionally, further research is needed to shed light on the intricate molecular mechanism underlying immune response to various types and intensity of exercise, one of the foremost being the impact of regular long-term moderate exercise in the general population.

The influence of different training programs exerted positive changes on motor and functional abilities, and the greatest improvement was observed in the circular training group. Likewise, several authors [46,47] investigated and concluded that resistance training has a positive effect on the locomotor system, maintaining functional abilities, osteoporosis prevention and management of lower back pain. It was concluded that a Nordic walking training program allows an increase in exercise intensity and adherence to a training program without increasing the perception of effort, leading to enhanced aerobic capacity [48]. Our research confirmed that two Nordic walking training sessions per week induced a decrease in body fat and weight parameters, following an increase in functional abilities. Furthermore, a combination of medium- to high-intensity exercise and resistance training appeared to have the greatest improvement in motor performance. These regimes also result in an improvement in the cardiovascular risk profile in overweight and obese participants compared to inactivity and the absence of exercise [49]. Therefore, a combination of exercise training should be recommended for overweight and obese adults.

## 5. Conclusions

To summarize, our results indicate the impact of medium- to high-intensity exercise in an overweight and obese, previously inactive, population resulting in an IgG glycan composition shift towards a pro-inflammatory glycan pattern. However, this is well expected given the characteristics of the study population and considering the fact that the participants were at the mere beginning of the exercise regime. On the other hand, three months of physical activity have already shown that there is a small but positive impact on cardiovascular risk profile, highlighting once again the importance of regular exercise on cardiovascular and overall health. It would be interesting to observe the impact of a longer exercise intervention in this type of population to better understand the exact role of IgG N-glycosylation changes and to further investigate the pro- and anti-inflammatory effects of physical exercise.

## Figures and Tables

**Figure 1 biomolecules-13-00762-f001:**
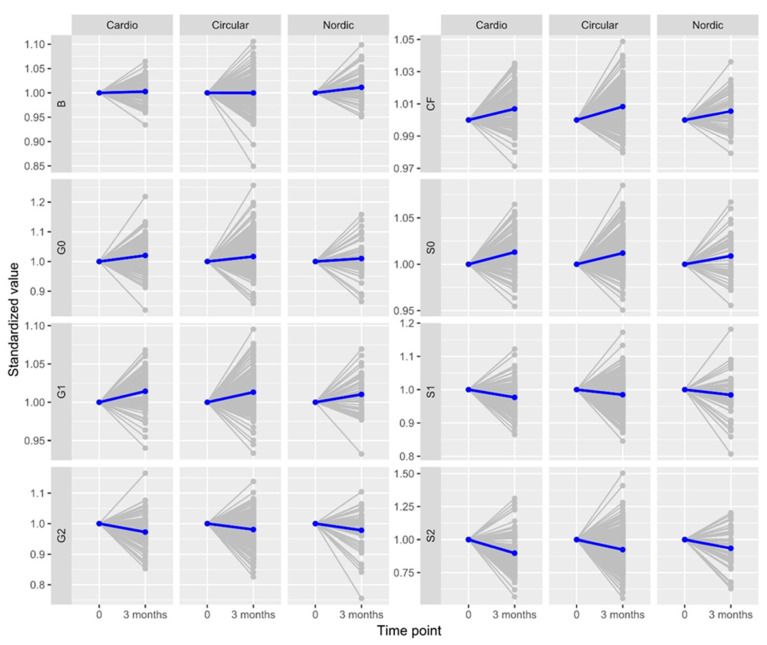
Changes in IgG glycome composition at two time points, normalized to the first point, and derived glycan traits induced by a 3-month exercise regime. The extent of change is expressed as fractions of standard deviation of a specific glycan trait. G0, agalactosylated glycans; G1, monogalctosylated glycan structures; G2, glycans with two galactoses; B, glycans with bisecting GlcNAc; S0, asialylated glycans; S1, glycans containing one sialic acid; S2, glycans containing two sialic acids; CF, glycans with core fucose.

**Figure 2 biomolecules-13-00762-f002:**
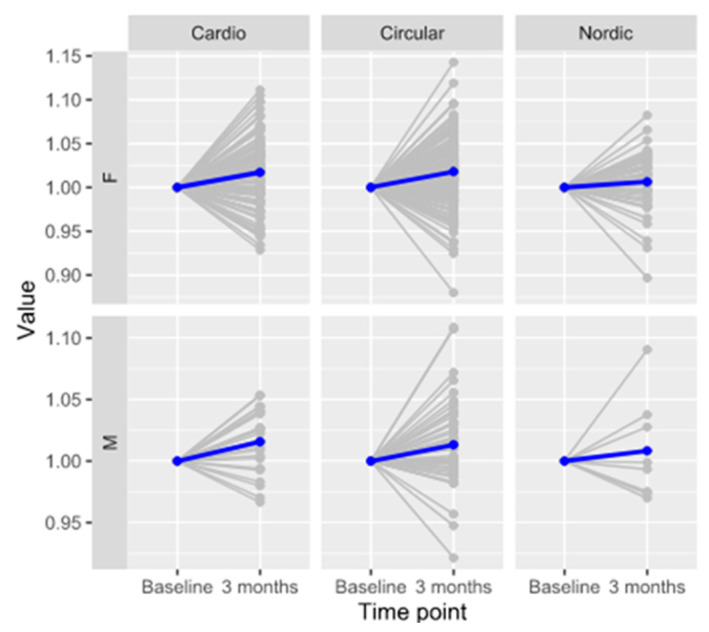
Changes in glycan structure FA2[3]G1 (GP9) at two time points for each training protocol, sex stratified and normalized to the first point; previously reported predictive biomarker for heart attack and stroke in women.

**Table 1 biomolecules-13-00762-t001:** Descriptive parameters and interactions (TIME*GROUP) in motor and functional abilities tests.

Variable	Circular Training		Cardio Training		Nordic Walking		Interaction TIME*GROUP		
	1	2	1	2	1	2			
	$\bar{x}\;\pm \;\mathrm{SD}$	$\bar{x}\;\pm \;\mathrm{SD}$	$\bar{x}\;\pm \;\mathrm{SD}$	$\bar{x}\;\pm \;\mathrm{SD}$	$\bar{x}\;\pm \;\mathrm{SD}$	$\bar{x}\;\pm \;\mathrm{SD}$	F	*p*	Partial ŋ2
Lunges	12.20 ± 3.22	16.36 ± 3.24	12.65 ± 4.00	15.95 ± 2.83	11.34 ± 2.20	15.57 ± 3.53	3.80	0.02 *	0.02
Wall sit	46.76 ± 27.10	76.20 ± 39.79	46.27 ± 34.02	76.33 ± 40.59	49.64 ± 24.26	74.12 ± 28.43	0.73	0.48	0.00
Squats	34.52 ± 9.45	46.59 ± 10.36	35.87 ± 9.01	44.86 ± 10.15	32.75 ± 7.81	46.08 ± 10.35	8.03	<0.01 *	0.04
Push-ups	23.25 ± 8.64	30.03 ± 8.96	21.30 ± 10.23	25.69 ± 9.86	23.68 ± 7.34	27.53 ± 7.48	7.22	<0.01 *	0.04
Plank	58.97 ± 32.57	118.59 ± 58.67	57.85 ± 28.54	106.09 ± 48.04	52.28 ± 21.15	86.34 ± 41.41	9.82	<0.01 *	0.05
Beep test	5.40 ± 2.13	6.85 ± 2.53	5.22 ± 1.64	6.38 ± 2.07	4.60 ± 1.38	5.92 ± 1.97	2.25	0.11	0.01

Legend: Variable—motor and functional ability tests; 1,2—initial and final testing; $\;\bar{x}$—average values; SD—standard deviation; F—f value; *p*—level of significance; partial ŋ2—measure of effect size; * marked values are significant when *p* < 0.05.

**Table 2 biomolecules-13-00762-t002:** Post hoc test for significant interactions.

**Lunges**								
Interaction	Program	Time	{1}	{2}	{3}	{4}	{5}	{6}
1	CARDIO	1		<0.01 *	0.82	<0.01 *	0.15	<0.01 *
2	CARDIO	2	<0.01 *		<0.01 *	0.90	<0.01 *	0.98
3	CIRCULAR	1	0.82	<0.01 *		<0.01 *	0.52	<0.01 *
4	CIRCULAR	2	<0.01 *	0.90	<0.01 *		<0.01 *	0.60
5	NORDIC	1	0.15	<0.01 *	0.52	<0.01 *		<0.01 *
6	NORDIC	2	<0.01 *	0.98	<0.01 *	0.60	<0.01 *	
**Squats**								
Interaction	Program	Time	{1}	{2}	{3}	{4}	{5}	{6}
1	CARDIO	1		<0.01 *	0.82	<0.01 *	0.38	<0.01 *
2	CARDIO	2	<0.01 *		<0.01 *	0.68	<0.01 *	0.98
3	CIRCULAR	1	0.82	<0.01 *		<0.01 *	0.84	<0.01 *
4	CIRCULAR	2	<0.01 *	0.68	<0.01 *		<0.01 *	1.00
5	NORDIC	1	0.38	<0.01 *	0.84	<0.01 *		<0.01 *
6	NORDIC	2	<0.01 *	0.98	<0.01 *	1.00	<0.01 *	
**Push-ups**								
Interaction	Program	Time	{1}	{2}	{3}	{4}	{5}	{6}
1	CARDIO	1		<0.01 *	0.52	<0.01 *	0.68	<0.01 *
2	CARDIO	2	<0.01 *		0.10	<0.01 *	0.66	0.89
3	CIRCULAR	1	0.52	0.10		<0.01 *	1.00	0.02 *
4	CIRCULAR	2	<0.01 *	<0.01 *	<0.01 *		<0.01 *	0.44
5	NORDIC	1	0.68	0.66	1.00	<0.01 *		<0.01 *
6	NORDIC	2	<0.01 *	0.89	0.02 *	0.44	<0.01 *	
**Plank**								
Interaction	Program	Time	{1}	{2}	{3}	{4}	{5}	{6}
1	CARDIO	1		<0.01 *	1.00	<0.01 *	0.97	<0.01 *
2	CARDIO	2	<0.01 *		<0.01 *	0.13	<0.01 *	0.07
3	CIRCULAR	1	1.00	<0.01 *		<0.01 *	0.92	<0.01 *
4	CIRCULAR	2	<0.01 *	0.13	<0.01 *		<0.01 *	<0.01 *
5	NORDIC	1	0.97	<0.01 *	0.92	<0.01 *		<0.01 *
6	NORDIC	2	<0.01 *	0.07	<0.01 *	<0.01 *	<0.01 *	

Legend: Interaction—number of interactions; program—cardio, circular and nordic training program; time—initial (1) and final testing (2); * marked values are significant when *p* < 0.05.

**Table 3 biomolecules-13-00762-t003:** Estimated effects of training intervention on levels of IgG N-glycan-derived traits. Effects are calculated relative to the baseline. Sample size per group (N), effect size (s.d.) and standard error (SE). *p*-values were adjusted for multiple testing using Benjamini–Hochberg procedure to control for false-discovery rate (FDR).

**CIRCULAR (*N* = 229)**			
Trait	Effect (s.d.)	SE	*p*-value (adjusted)
G0	0.040	0.015	1.12 × 10^−2^
G1	0.258	0.028	1.55 × 10^−16^
G2	−0.094	0.019	2.13 × 10^−6^
B	−0.026	0.016	1.32 × 10^−1^
CF	0.523	0.050	1.07 × 10^−20^
S0	0.258	0.032	7.01 × 10^−14^
S1	−0.085	0.020	5.63 × 10^−5^
S2	−0.500	0.058	3.06 × 10^−15^
**CARDIO (*N* = 115)**			
Trait	Effect (s.d.)	SE	*p*-value (adjusted)
G0	0.067	0.026	1.61 × 10^−2^
G1	0.316	0.047	1.65 × 10^−9^
G2	−0.151	0.030	2.73 × 10^−6^
B	0.006	0.021	7.93 × 10^−1^
CF	0.521	0.088	6.55 × 10^−8^
S0	0.277	0.044	1.17 × 10^−8^
S1	−0.128	0.027	8.74 × 10^−6^
S2	−0.628	0.076	9.03 × 10^−13^
**NORDIC (*N* = 53)**			
Trait	Effect (s.d.)	SE	*p*-value (adjusted)
G0	0.002	0.034	9.52 × 10^−1^
G1	0.186	0.058	3.56 × 10^−3^
G2	−0.126	0.050	1.86 × 10^−2^
B	0.086	0.041	5.39 × 10^−2^
CF	0.406	0.118	1.77 × 10^−3^
S0	0.181	0.075	2.54 × 10^−2^
S1	−0.075	0.054	1.89 × 10^−1^
S2	−0.445	0.124	1.16 × 10^−3^

**Table 4 biomolecules-13-00762-t004:** Effect size (s.d.), standard error (SE), sample size per group (N) and *p*-values for testing the effect of exercise on GP9 (age-adjusted) levels in IgG glycome. False discovery rate was controlled using Benjamini–Hochberg procedure.

Program	Sex	Effect (s.d.)	SE	*p* (Adjusted)	N
Cardio	M	0.260	0.087	9.26 × 10^−3^	24
	F	0.139	0.037	4.20 × 10^−4^	91
Circular	M	0.124	0.055	3.53 × 10^−2^	54
	F	0.151	0.022	2.64 × 10^−10^	177
Nordic walking	M	0.053	0.299	8.86 × 10^−1^	8
	F	0.058	0.043	2.01 × 10^−1^	45

## Data Availability

The data presented in this study are available on request from the corresponding author. The data are not publicly available due to privacy or ethical restrictions.

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
