# Peer review of "Physical Exercise Induces Significant Changes in Immunoglobulin G N-Glycan Composition in a Previously Inactive, Overweight Population"

_biomolecules, 2023, doi:10.3390/biom13050762_

Round 1

Reviewer 1 Report

Abstract

When describing the research group in the abstract, it is said that it was a group of obese people, and according to the BMI, obesity is from level 30, which with a variance of 4.81 means that it could also be overweight people. I recommend bringing it into line.

Participants
It is said here that: Physical inactivity was determined by a questionnaire on the general state of health and related to the last 5 years, during which the respondents were not actively involved in any form of recreational activities. I would welcome a more detailed description of whether and how possible comorbidities were taken into account when dividing into advised exercise groups and whether any exclusive criteria were applied when compiling the research group.

Discussion

It follows from the description that there were both women and men in the group, but in the discussion you point to a positive effect in the category of women, demonstrated in a similar study (line 333), which, of course, is problematically transferable if your group is not divided into men and women.

Author Response

Please find document attached.

Reviewer 2 Report

In this manuscript, the authors showed different IgG glycosylation profiles after exercise. Given the importance of IgG glycans in many biological processes, it is an interesting question. However, the authors make big claims that are not supported by the results.

Starting with the title, for example. It implies causality for things that were not measured. There is no measure of inflammation or CVD status. In the discussion, many assumptions were made, such as the core fucose in the participants is increasing; therefore, there is a change in ADCC that can prove the post-exercise immune function depression theory. From all this sentence, the only thing measured is fucosylation; there is no ADCC and no immune function test. Also, in the last part of the discussion, the authors stated "positive impact on cardiovascular risk profile"; however, one single glycan can not and does not represent a very complex disease like CVD. CVD is an umbrella term for many diseases, such as high blood pressure, stroke, coronary heart disease, etc. What about the other glycans with a much bigger difference and a very strong p-value?

For making such big claims, it would be great to actually measure at least pro-inflammatory cytokines IL6 and TNFa to check for a correlation between the glycans and the inflammation. 

The tone of all the manuscripts has to be drastically decreased to reduce many speculations and implications. It is a good study, but it is a descriptive study without any mechanism. 

Minor comments:

  • The explanation for V02 max comes only in the second paragraph of the introduction, but the first paragraph already talks about it. It would be good to bring it to the first paragraph.
  • Add definitions of sd and se to tables
  • some words miss a space between them

Author Response

Please find document attached.

Round 2

Reviewer 2 Report

The authors have addressed all my concerns.